# Immobilization of Cadmium by Molecular Sieve and Wollastonite Is Soil pH and Organic Matter Dependent

**DOI:** 10.3390/ijerph18105128

**Published:** 2021-05-12

**Authors:** Meiliang Dong, Rong Huang, Peng Mao, Long Lei, Yongxing Li, Yingwen Li, Hanping Xia, Zhian Li, Ping Zhuang

**Affiliations:** 1Key Laboratory of Vegetation Restoration and Management of Degraded Ecosystems, South China Botanical Garden, Chinese Academy of Sciences, Guangzhou 510650, China; mldong2017@scbg.ac.cn (M.D.); huangrong1992@163.com (R.H.); pmao@scbg.ac.cn (P.M.); leilong19@mails.ucas.ac.cn (L.L.); liyongxing.7@163.com (Y.L.); liyw@scbg.ac.cn (Y.L.); xiahanp@scbg.ac.cn (H.X.); 2University of Chinese Academy of Sciences, Beijing 100049, China; 3Southern Marine Science and Engineering Guangdong Laboratory (Guangzhou), Guangzhou 511458, China

**Keywords:** immobilization, cadmium, wollastonite, molecular sieve, organic matter, competitive adsorption

## Abstract

The excessive cadmium (Cd) concentration in agricultural products has become a major public concern in China in recent years. In this study, two amendments, 4A molecular sieve (MS) and wollastonite (WS), were evaluated for their potential passivation in reducing Cd uptake by amaranth (*Amaranthus tricolor* L.) in six soils with different properties. Results showed that the responses of amaranth biomass to these amendments were soil-property-dependent. The effects of MS and WS on soil available Cd were in turn dependent on soil and amendment properties. The application of WS and MS at a dose of 660 mg·kg^−1^ Si produced the optimum effect on inhibiting Cd accumulation in amaranth shoots (36% and 34%, respectively) and did not affect crop yield. This was predominantly attributed to the marked increase in pH and exogenous Ca or Na, which facilitated the adsorption, precipitation, and complexation of Cd in soils. The immobilization effects of WS and MS were dependent on soil properties, where soil organic matter may have played an important role. In conclusion, MS and WS possess great potential for the remediation of Cd-contaminated acidic soils.

## 1. Introduction

Recently, the behavior of humans has resulted in the deterioration of the global ecological environment. Heavy metal contamination is a major public concern across the world [1]. Cadmium (Cd) exposure among Chinese citizens has steadily increased through soil–food transfer in the last two decades [2]. According to the national bulletin of the Soil Pollution Survey of China [3], 19.4% of the arable land survey sites fail to attain soil environment quality standards, 7.0% of the survey sites containing a higher Cd concentration than the permissible limit. Due to excessive nitrogen fertilization, as well as prolonged soil desilicification and allitization, farmland in Southern China has become acidified [4,5]. This has contributed to an increase in Cd mobility and availability, resulting in higher Cd uptake in plants [6]. Cadmium is a toxic element, which can cause damage to the kidneys, bones, and lungs [2,7]. It is therefore vital to investigate sustainable strategies to modulate Cd availability in contaminated soil and improve food safety standards.

In situ immobilization has been recognized as a cost-effective, simple, and efficient method to cope with heavy metal contamination in soil, especially on large-scale farmland [8,9]. While several immobilization amendments, such as phosphate compounds, liming materials, organic compost, metal oxides, and biochar, have been effective in decreasing the bioavailability, mobility, and toxicity of heavy metals in soils [10], as well as in reducing crop heavy metal accumulation, their efficiency has varied significantly. For instance, sepiolite tested in Zhejiang Province, China could reduce rice grain Cd concentration by 47–49% [11], compared with 79–83% [12] and 17–60% [13] in Hunan Province. These previous studies indicated that amendments were sensitive to the soil, as the behavior of heavy metals was affected by the varied properties among different soils.

The effects of soil amendments on heavy metals could be dependent on soil properties, as the latter can potentially influence changes in the soil pH or in the metal complexation process. Previous studies have demonstrated that acidic soils with relatively low pH values [14], light contamination [15], and poor Cd adsorption capacity [16] were suitable for immobilization remediation processes. As alkaline amendments can notably increase soil pH and soil adsorption on site, as well as promote metal adsorption and complexation, they can, therefore, reduce the mobility of metals in the soil. Soil cation exchange capacity (CEC) is one of the determinants of heavy metal sorption in soil, and of bioavailability [17,18]. It is possible that soil organic matter, clay particles, as well as Fe and Al oxides, which determine CEC, can attract metal cations by electrostatic adsorption because of their negatively charged surfaces [19]. Therefore, a high CEC can increase heavy metal ion retention in the soil, thereby decreasing metal availability [20]. Organic matter can either enhance or restrict heavy metals’ mobility [20], which are released or detained owing to their different compositions and concentrations. Unfortunately, the role of soil properties in heavy metal behavior and amendment performance is not yet properly understood.

In our previous studies, 4A molecular sieve (MS) and Wollastonite (WS) exhibited effective immobilization in Cd concentrations in grain amaranths by 68–89% [21] and by 67–71% in amaranth [22]. These two alkaline materials effectively reduced heavy metal mobility and bioavailability in acidic soil [23]. The 4A molecular sieve (MS) is a synthetic silicoaluminate material with a nanoscale microporous cubic lattice, which can simultaneously increase soil pH and adsorb heavy metals [24,25]. Wollastonite (WS) is a naturally occurring, low-cost, and environmentally friendly calcium inosilicate mineral (Ca_3_[Si_3_O_9_]), which can also increase soil pH. The detailed properties of MS and WS were reported in our previous study [6] and are presented in Appendix A. Some amendments have immobilization effects on heavy metals in polluted soils, which can ensure the safety of agricultural production. However, different soil properties may lead to different effects of amendments. Therefore, research on the effect of amendments used in a variety of soils with different properties is conducive to the correct use of amendments. Owing to the immobilization effects of heavy metals, the above two materials were investigated to determine whether suppression of crop Cd uptake by amendments was dependent on the properties of the soil in Southern China. A leafy vegetable, amaranth (*Amaranthus tricolor* L.), being a food source with a high health risk due to Cd, was selected and grown in six soils with different properties amended with two alkaline materials. The main objective of this study was (1) to study changes in immobilization of Cd by two amendments under different soil properties and (2) to investigate the possible mechanism of metal adsorption by different amendments and its field applicability.

## 2. Materials and Methods

### 2.1. Soil Samples and Amendments

Six Cd-contaminated soils (soils 1–6) were sampled from different sites of agricultural land (Appendix A) in Shaoguan, Guangdong, China (23°53′–25°31′ N, 112°5′–114°45′ E). This region has a humid subtropical climate with an annual average rainfall of 1640 mm and temperature of 20.1 °C. Soils 1 and 2 were collected from a farmyard and a cropland, respectively, close to a smelting plant in Dongtang, Renhua county. Soil 3 was sampled from farmland in Maba, Qujiang county, approximately 4 km from a smelting plant. Soil 4 was collected from a paddy field contaminated by emissions from coal burning, located approximately 8 km from a thermal power plant in Zhangshi, Qujiang county. Soil 5 was obtained from farmland repeatedly irrigated with polluted water from the Dabaoshan Mine in Wengyuan county, and soil 6 was sampled from the nutrient-rich soil covering amended mineral slag in Renhua county. Surface soil was sampled at a depth of 20 cm from each site using a five-point sampling method. Soils were air-dried and passed through a 10 mm sieve for pot-based experiments, and through a 0.84 mm or 0.25 mm sieve for chemical and Cd concentration analyses. The basic chemical properties of sampled soils are shown in Table 1. Soil pH values ranged from 4.90 to 7.77, while soil CEC ranged from 0.4 to 10.3 cmol(+) kg^−1^.

The amendments, i.e., MS and WS, were both commercially available products. MS is a synthetic sodium-silicoaluminate with a nanoscale microporous cubic lattice. It is usually used as a sorbent in water and can adsorb molecules with a critical diameter less than 4A (i.e., <0.4 nm). WS is a natural calcium inosilicate mineral. The characteristics of the two soil amendments are shown in Appendix A.

### 2.2. Pot-Based Experiments

Experiment A: pot-based experiments were established in the open field of the South China Botanical Garden, Chinese Academy of Sciences, Guangzhou, in order to explore the effects of soil properties on Cd immobilization. Each plastic pot (201 mm × 129 mm × 140 mm) contained 3.0 kg of Cd-contaminated soil. Twenty-four treatments, each with four replicates, were designed: WS and MS were applied to six soils (soils 1–6) at dosages of 1.9 g·kg^−1^ soil and 2.08 g·kg^−1^ soil, respectively, equivalent to Si applied at 440 mg·kg^−1^ soil. Soils with no amendments were used as the control. Amendments were thoroughly manually mixed with the soils, and the mixtures incubated for 30 d, with water content maintained at approximately 70%. Urea, ammonium dihydrogen phosphate, and potassium sulphate were added to this mixture as basic fertilizers, with the concentrations of N at 0.2 g·kg^−1^ soil, P_2_O_5_ at 0.15 g·kg^−1^ soil, and K_2_O at 0.2 g·kg^−1^ soil. Pots containing the fertilized soils were further incubated for 3 d. Three amaranth (*Amaranthus tricolor* L.) seedlings were then transplanted to each pot after 15 d in culture. Plant growth was continued for 31 d (from 19 October to 19 November 2018). The final amaranth biomass production was determined by assessing the corresponding fresh and dry mass after a growth period of 45 d.

Experiment B: to determine the optimum application rate of soil amendment for metal immobilization, dosage tests were set up. The two amendments were applied to Maba soil (soil 3) at dosages corresponding to Si concentrations of 0, 220, 440, 660, and 880 mg·kg^−1^ soil, which were equivalent to WS at 0.95, 1.9, 2.85, and 3.8 g·kg^−1^ soil, and MS at 1.04, 2.08, 3.12, and 4.16 g·kg^−1^ soil. Incubation, planting, and management conditions were maintained as for experiment A.

### 2.3. Plant and Soil Sample Analyses

Plants were harvested, thoroughly rinsed with deionized water, and separated into shoots and roots. The dry weight of plant tissues was recorded once a constant weight was obtained in the oven at 70 °C. Dried plant samples were ground and passed through a 0.5 mm sieve for chemical analysis.

Following amaranth harvest, soil samples were collected, air-dried, and passed through either a 2 mm or a 0.15 mm sieve.

To determine total Cd content, soil and plant samples were digested in a microwave oven with a HNO_3_-HF-HCl mixture (6:2:3; v:v:v) and 5 mL concentrated HNO_3_, respectively (Standard codes HJ 832-2017 and GB 5009.268-2016, China). The digested mixtures were diluted to 25 mL with 1% HNO_3_, filtered, and the Cd concentrations in both soil and plant samples were determined by inductively coupled plasma emission spectrometry (ICP-OES; Optima 2000 DV, PE, USA).

Soil pH was measured in 1:2.5 soil to water ratio suspensions using a pH meter (Mettler Toledo FE-20K). The CEC of soils was detected using the hexamminecobalt trichloride solution–spectrophotometric method, according to the national environmental protection standards of China (Code HJ 889-2017). In short, soil samples (3.5 g) were percolated with 50 mL hexamminecobalt trichloride solution and shaken for 1 h. Concentration of hexamminecobalt trichloride was measured by a UV–Vis Spectrophotometer (UV-2000, UNICO Instrument Co., Ltd., Shanghai, China). The Olsen method was used to measure available P in soils (pH < 6.5), using 0.5 M NaHCO_3_ (pH 8.5, 1:20 soil: NaHCO_3_ solution ratio) as extractant. When pH > 6.5, soils were extracted with a 0.03 M NH4F/0.025 M HCl solution (1:10 soil: solution ratio). Suspensions were shaken for 0.5 h, and available P was determined using the molybdenum blue spectrophotometric method (Lu, 2000). Soil organic matter was analyzed using the wet digestion method with K_2_Cr_2_O_7_/H_2_SO_4_, followed by titration with FeSO_4_. Available soil Cd and Ca were measured by DTPA extraction and using Mehlich-3 solution, respectively, following the methods described by Mao et al. [6]. Available soil Na was extracted with 1 M NH4OAc and suspensions were shaken for 1 h [26]. Changes in Cd speciation in soils were measured using the sequential extraction method reported by Tessier et al. [27], and the operation is shown in Appendix A. Available Ca, Na, and Cd, as well as Cd speciation, in soil were analyzed by atomic absorption spectroscopy (AAS, contrAA 800, Analytik Jena, Germany), and the DTPA extracts were determined within 48 h. The reduced percentage of available Cd represented the effect of soil amendments. The immobilization rate of Cd (Cd-IR) was determined as follows [28]:(1)$$\mathrm{Immobilization}\;\mathrm{rate}\;(\%)=\;\;\frac{\mathrm{Available}\;\mathrm{Cd}\;\mathrm{in}\;\mathrm{control}-\mathrm{Available}\;\mathrm{Cd}\;\mathrm{in}\;\mathrm{amened}\;\mathrm{soil})}{\mathrm{Available}\;\mathrm{Cd}\;\mathrm{in}\;\mathrm{control}}\;\times \;100\;$$

All samples were measured in reagent blanks. Reference rice sample GBW (E) 100357 and soil sample GBW07437 were employed, with Cd recovery rates ranging from 93% to 102% and 94% to 106%, respectively. ICP multi-element standard solution (GNM-M27195-2013) was used to calibrate the concentrations of Cd and Ca. The solution was also measured immediately after calibration, after every 20 samples, and at the end of the analysis process, used as the quality control (QC) standard with a recovery limit of 100% ± 10%.

### 2.4. Statistical Analyses

All data presented in tables and figures are means ± SE of four replicates. One-way ANOVA was employed to determine significant (*p* < 0.05) differences among treatments, and data obtained were subjected to Tukey’s honestly significant difference test in order to further highlight these differences (*p* < 0.05). Pearson’s correlation coefficients were calculated to detect correlations (at a confidence interval of 95%, using a two-tailed test) among soil chemical properties, soil available cadmium contents, and plant cadmium contents.

All statistical analyses were performed by SPSS 19.0. Origin 9.0 software ((OriginLab Corporation, Northampton, MA, USA)) was used to produce all the figures.

## 3. Results

### 3.1. Amaranth Biomass and Cd Accumulation

Amaranth biomass and the response of biomass to amendments were soil-property-dependent. As shown in Figure 1A, amaranth grown in soil 3 achieved a much higher biomass than when cultivated in other soils. The shoot biomass of amaranth grown in unamended soil 3 was 5.45 g per pot, while amaranth yield in other unamended soils was less than 3 g per pot, with the lowest being in soil 2 at only 0.99 g per pot. In soils 1 and 5, the amendments produced a negative effect on shoot biomass. Amendments MS and WS decreased the shoot biomass by 15% and 48%, respectively, in soil 1, and significantly reduced shoot biomass by 49% and 32%, respectively, in soil 5. By contrast, soil amendments produced a positive effect on shoot biomass in soils 4 and 6. MS and WS increased shoot biomass by 45% and 13%, respectively, in soil 4, and by 56% and 104%, respectively, in soil 6. In soils 2 and 3, WS increased the shoot biomass by 118% and 21%, respectively, compared with the control, while treatment with MS produced no difference when compared with the control. Amaranth biomass varied with the addition of different dosages of soil amendments (Figure 1B). A low Si dose (220 mg·kg^−1^) resulted in a biomass reduction, with a decrease of 19% for MS and 13% for WS. A dose of Si which was too high (880 mg·kg^−1^) also decreased amaranth biomass, with a reduction of 21% for MS and 34% for WS. However, specific doses of either amendment could markedly increase biomass. Si addition of 440 mg·kg^−1^ of MS to soil increased amaranth biomass by 20%, while 660 mg·kg^−1^ Si of added WS increased biomass by 21%.

Amaranth accumulated varying levels of Cd from different unamended soils (Figure 2A). Cd concentration in amaranth shoots was 30.8 mg·kg^−1^ in soil 3, but only 13.2 mg·kg^−1^ in soil 1 and 18.0 mg·kg^−1^ in soil 4. For soil 6 (with Cd at the extremely high concentration of 20.8 mg·kg^−1^), accumulated Cd in amaranth shoots was 18.3 mg·kg^−1^, at a much lower concentration than in soil 3. While amendments inhibited Cd uptake in amaranth, the extent of inhibition was dependent on the soil property. Both amendments significantly (*p* < 0.05) decreased Cd accumulation in soils 4 and 5, with an approximate reduction of 55% and 53–68%, respectively. In soils 1, 2, and 3, both amendments reduced Cd content by only 13–30%. Almost no amendment effect was observed in soils 3 and 6. Furthermore, no significant (*p* > 0.05) difference was observed between the effects of MS and WS. However, in the case of soil 3, an increasing amendment dose significantly (*p* < 0.05) linearly decreased Cd accumulation in amaranth shoots (Figure 2B). The highest Si dose (880 mg·kg^−1^) resulted in a 52% drop in amaranth Cd concentration by WS, or in a 42% drop by MS. The two amendments performed very similarly, as shown by the two lines nearly overlapping each other.

### 3.2. Changes in Soil Properties

The addition of MS and WS significantly (*p* < 0.05) increased the pH and CEC of all soils, except soil 6 (Table 2). The pH of soil 3, in particular, presented a tendency to rise with increasing addition rates of the two amendments (Table 3). While no significant difference was observed in the soil CEC among treatments with different MS doses, the increase in the amount of WS led to a raised soil CEC. Organic matter (OM) contents were found to be 9.6–39.7 g·kg^−1^. Six soils contained 0.38–20.8 mg·kg^−1^ of Cd, representing different levels of Cd contamination in the agricultural soil of Southern China. Available Cd in the six selected soils varied from 0.15 to 2.02 mg·kg^−1^. These sampled soils also contained 136–1231 mg·kg^−1^ of available Ca, 46.4–152 mg·kg^−1^ of available K, and 11–62.7 mg·kg^−1^ of available Na. The concentrations of available Na and Ca were significantly (*p* < 0.05) increased with the addition of MS and WS, respectively (Table 3). Results showed that the concentrations of Mg and K increased or decreased after WS and MS were applied, depending on different properties of soil (Appendix A). WS and MS addition significantly (*p* < 0.01) decreased available Zn and Mn concentrations in soils, with a reduction of approximately 90% for available Zn (Appendix A).

### 3.3. Changes in Soil Cd Availability and Fraction Distribution

Soil amendments did not necessarily produce a decline in Cd availability. There was no difference in reduced Cd availability in soils between the two amendments (Figure 3A). In soils 1, 4, and 5, the immobilization rate of available Cd for the two amendments was 10–18% (Figure 3B). However, both tested amendments (at a dose of 440 mg·kg^−1^ Si) slightly increased soil available Cd concentrations in soils 2, 3, and 6, with the highest negative immobilization rate of available Cd (−11%) being in soil 3 (Figure 3B). Available Cd concentrations initially increased at 220 mg·kg^−1^ Si but further declined with increasing addition dosages of the two amendments (Figure 3C). In particular, soil available Cd concentrations were higher in amendment-treated than in untreated soil, resulting in an immobilization rate of available Cd below 0 for WS treatment (Figure 3D). In a similar pattern, a reduction in available Cd was obtained when MS was added at the high Si dose of 880 mg·kg^−1^.

Considerable variation was observed in Cd distribution among different fractions of the six soils, based on the results of sequential extraction. As shown in Figure 4A, both amendments decreased the proportions of exchangeable Cd (F1) in soils 1, 4, and 5. Conversely, the proportions of exchangeable Cd (F1) in soils 2, 3, and 6 remained unchanged or even increased slightly under the treatments. Generally, the proportions of exchangeable Cd (F1) declined with increasing addition rates of the two amendments (Figure 4B). However, exchangeable Cd (67–72%) in soils treated with 220 mg·kg^−1^ of MS, as well as with all the WS treatments, were still higher than the control (67%), while the proportions of Cd in the F2, F3, and F4 forms were lower than that of the control, suggesting that the application of WS and a low dose of MS remobilize unavailable Cd (F2, F3, and F4 forms) into its available speciation (F1 form).

### 3.4. Interactions between Cd Accumulation and Soil Cd Availability with Soil Properties

Correlation analyses between soil chemical properties and Cd accumulation in amaranth shoots, as well as soil Cd availability, are presented in Table 4. Data obtained (Table 4A) showed that Cd accumulation in amaranth shoots was significantly affected by OM (*r* = 0.825), CEC (*r* = 0.314), available Cd (*r* = 0.768), and available Na (*r* = −0.305). There were significant (*p* < 0.01) positive correlations between soil Cd availability and OM (*r* = 0.775), CEC (*r* = 0.708), available Ca (*r* = 0.473), and available P (*r* = 0.299). With regard to soil 3 treated with MS and WS at different application dosages (Table 4B), Cd concentrations in shoots were significantly (*p* < 0.01) and negatively correlated to pH (*r* = −0.853) and available Ca (*r* = −0.462) in the soil. The available Cd in soil 3 displayed a positive correlation with OM (*r* = 0.402) but negative correlations with soil pH (*r* = −0.423) and available Na (*r* = −0.547).

To determine the relationship between pH, OM, CEC, Ca availability, Na availability, Cd availability, and Cd accumulation in amaranth shoots, and to identify the factors affecting the immobilization and uptake of Cd, principal component analysis (PCA) was carried out based on the characteristics of the different properties of soil and treatments (Figure 5). Results showed that two principal components could explain 71% of the total variation in original variables. Soil available Ca and CEC were close to soil available Cd, indicating positive relations between them. PC1 can be interpreted as the soil integrated properties that related to inhibiting the immobilization of soil available Cd. Organic matter was closest to the Cd concentration of amaranth shoot, denoting a significant positive relation. Soil pH was negatively related to the Cd concentration of amaranth shoot and positively related to soil available Na. PC2 can be interpreted as the soil integrated properties that are related to enhancing amaranth Cd uptake.

## 4. Discussion

### 4.1. Plant Biomass and Metal Accumulation Were Dependent on Properties of Soil and Amendments

Soil productivity for amaranth growth varied markedly, as shown in Figure 1A. Soil properties, including soil pH, organic matter, and available P, were potential key factors affecting soil productivity, as their effects were interactive and complex [29]. Soil 3 exhibited high productivity, with amaranth biomass from unamended soil being 5.5 times higher than that of soil 2. Soil amendments for heavy metal immobilization either largely increased or decreased crop biomass, depending on the soil and amendment properties [14,30]. In the present study, the application of WS produced positive effects on plants growing in soils 2, 3, and 6, which was consistent with our previous study [6]. This is most likely due to the mitigation effects of Si on toxic metal stress, which reflect a certain regulatory effect on plant growth [31,32]. However, some inconsistencies were also detected between results of the present study (soils 1 and 5) and those of our previous study [22], which highlighted a few obvious effects of WS application on plant biomass. The present results showed that plant biomass was dependent on the properties of soil and amendment used. The change in amaranth biomass observed with different application dosages indicated that the quantity of amendment used was an important factor in this process (Figure 1B). A decline in growth could occur due to the dose being too low or too high, indicating amendments each performed at a different optimal dose in soil with different properties. Amaranth biomass markedly increased and reached a maximum value at the Si addition dosages of 440 mg·kg^−1^ for MS and 660 mg·kg^−1^ for WS. This result was in agreement with a study reported by Lu et al. [21]. A soil-specific dosage test is therefore necessary for each amendment before its application.

The level of Cd uptake by amaranth cultivated in various soils was noticeably different (Figure 2A). The highest Cd concentration in amaranth grown in unamended soil was obtained in soil 3 and was 7.6 times higher than the lowest concentration obtained in soil 5. This was in agreement with the study by Liu et al. [33], which showed the varying levels of Cd uptake by wheat grown in 18 different soils. In the present study, the suppression of crop Cd accumulation by two amendments was different in various soils, with reductions of more than 55% in soils 4 and 5, and 13–30% in soils 1, 2, and 3. The highest Si application dosage (880 mg·kg^−1^) of the two tested amendments resulted in an approximately 50% reduction in crop Cd uptake (Figure 2B), while also sharply decreasing crop yield (Figure 1B). The suppression of Cd accumulation in amaranth plants was not due to decreased levels of available Cd, but more likely due to the deposit of silicon in plants, which potentially led to cell wall thickening of the epidermis, endoderm (Casparian strip), and vascular column, thereby limiting Cd translocation by the root apoplast [34,35]. The application of WS and MS in soil not only affected Cd absorption but also that of several mineral elements (Appendix A). A progressive reduction in Zn and Mn concentrations in amaranth shoots and roots revealed that these elements were simultaneously immobilized by WS and MS, similar to the findings of Houben et al. [36] and Wu et al. [22]. By contrast, owing to the combination effect of Si and Ca, elevated concentrations of Ca^2+^ from WS occur through ion competition and enter plants affected by Si [37]. Similarly, Na concentrations in amaranth were enhanced with increasing doses of MS, possibly owing to a large amount of available Na^+^ in the soil, accumulated by plants through ion exchange. The underlying process responsible for the accumulation of large amounts of Na by amaranth exposed to MS should be further explored.

### 4.2. Relationship between Cd Availability, Immobilization Rate, and Soil Properties

Soil pH has also been recognized as a critical factor affecting Cd availability in soils [38]. Earlier studies have reported that alkaline amendments decreased soil Cd availability by increasing soil pH [6,22]. Similar findings from the present study have confirmed that the application of both MS and WS increased soil pH (Table 2 and Table 3), leading to the deprotonation of variable charge functional groups from soil inorganic minerals and organic components, which increased the negative charge on the soil colloid surface, thereby promoting the electrostatic adsorption of Cd^2+^ into the soil [39]. Increasing soil pH also strengthened the specific adsorption of Cd^2+^ by soil, via promoting the transformation of free Cd^2+^ into precipitate, complex, iron and manganese oxides, as well as other stable fractions [40,41]. The present results showed that the addition of WS and MS can convert the available Cd fraction into its less available forms (e.g., carbonates, Fe-Mn oxides, or oxidizable fractions; Figure 4). As the pH of soil 6 was relatively high at 7.77 (Table 1), the application of alkaline amendments had no significant impact on its pH (Table 2) and Cd availability (Figure 3A). The negative correlation between soil pH and concentration of soil available Cd (Table 4B) also supported this conclusion. Due to the increase in the negative charge on the soil surface, the increase in soil pH may have led to the precipitation and adsorption of Cd in the soil, reduced Cd solubility and mobility in the soil, and thus reduced Cd accumulation in the plant. These results suggested that increasing soil pH to promote processes including surface adsorption, precipitation, and complexation was one of the main mechanisms responsible for Cd immobilization in acidic soils, but that a similar increase in pH did not improve alkaline soils for Cd immobilization.

The adsorption or binding of Cd and OM was unstable, with soil biological disturbance causing the release of dissolved OM and providing bound available Cd [42]. It should be noted that soil available Cd was positively correlated with soil OM content in the present study (Table 4). Similar results reported by Xu et al. [43] showed that the Cd availability of river sediments was positively correlated with the oxidizable Cd (Cd bounded to OM) extracted using the modified BCR sequential extraction method. Kumar [44] suggested that degradation of OM and age-related changes in functional groups produced heavy metal remobilization. This conclusion was also supported by the results of Cd speciation distribution in soil based on the Tessier sequential extraction method (Figure 4B). The proportion of oxidizable Cd (F4) decreased and exchangeable Cd (F1) increased under WS treatment, when compared to the control. This result verified that the release of Cd bound to OM contributed to the increase in soil exchangeable Cd. Although the available Cd in soils 2 and 3 was comparable to that of soil 1, their corresponding Cd immobilization rates were much lower than that of soil 1 (Figure 3B) due to the high soil OM content.

Figure 5 and Table 4A also indicate that soil CEC was positively correlated with soil available Cd. OM, as the main sorbent in soil [45], exhibited a significant positive correlation with CEC in this case (*r* = 0.344 **, *p* < 0.01; Figure 5). The reason for the positive correlation between OM and available Cd may also explain why CEC was positively correlated with soil available Cd. However, a negative correlation between soil CEC and soil Cd mobility was also reported by Markovic et al. [16] and Karak et al. [39]. The function of OM should be considered when planning and designing soil remediation strategies. The effects of soil CEC on Cd immobilization and adsorption require further study.

According to PCA results, the dominant influencing factors that correlated with soil available Cd were soil available Ca and soil CEC (Figure 5). Soil available Cd was closest to soil available Ca, consistent with data presented in Table 4A, indicating a significant positive correlation between available Cd and available Ca. This conclusion was also supported by the results of the batch sorption experiments (Appendix A), which showed that the Cd adsorption capacity of soil decreased significantly with an increase in the Ca: Cd molar concentration ratio, with the lowest Cd adsorption capacity being in WS-amended soil, owing to extra Ca introduced by WS. Ca competed with Cd for adsorption sites due to its stronger affinity with soil and also due to the stronger electric double layer compression of the bivalent electrolyte [46], thus decreasing soil Cd retention and increasing Cd mobility [47].

To verify the interaction of soil pH and OM with Cd immobilization rate, response surface plots were conducted to show the interactive effects on soil properties (soil pH and OM) on immobilization rate or soil Cd and reduction rate of shoot Cd by WS and MS. The interactions of soil pH (*x*-axis), soil OM (*y*-axis), and immobilization rate or reduction rate of Cd at the *z*-axis are presented in Figure 6. The elevated Cd immobilization rate in six soils as a result of the decrease in the content of soil OM was evaluated, resulting in the higher Cd reduction rate of amaranth. Together with an increase in soil OM from 9.6 to 39.7, there was a decrease in Cd immobilization rate from 10–18% to −11%. The lowest values of immobilization rate of Cd were observed in the lower right corners (pH-OM) of the plots, which corresponds with higher soil pH. For six soils, a lower soil pH value brought about the higher Cd reduction rate of amaranth.

### 4.3. Possible Mechanism for Cd Immobilization Using MS and WS

The present results showed that the application of the two tested amendments produced no change in available Cd in soils 2, 3, and 6 (Figure 3A), indicating that MS and WS do not necessarily suppress Cd uptake by lowering Cd availability in soil. In this study, the application of MS and WS at 220 mg·kg^−1^ Si increased Cd mobility and availability (Figure 3C and Figure 4B). This was not in line with the existing literature [40,48], where immobilization meant a reduction in the fraction of available Cd, as well as a decrease in Cd mobility. When WS was applied at a dose of 660 or 880 mg·kg^−1^ Si, the immobilization rate below 0 (Figure 3D) and the significant reduction in Cd concentration (Figure 2B) in amaranth shoots imply that other mechanisms could be involved in the suppression of Cd accumulation. Although both MS and WS provided a considerable proportion of Si, they did not contribute to the suppression of plant Cd accumulation, as confirmed in our earlier study [49], and were therefore not analyzed in detail here. In the case of WS, a Ca silicate mineral, Cd immobilization was widely associated with the addition of a large amount of free Ca ions from WS (Table 2 and Table 3). Exogenous Ca can alleviate Cd toxicity stress by reducing cell-surface negativity and competing for Cd^2+^ ion influx in WS treatment, resulting in a decrease in Cd uptake [50]. Li et al. [41] also reported that Ca-bearing amendments reduced Cd uptake in rice in paddy soil. As a typical chemical analogue of Cd, Ca competed with Cd for soil adsorption sites [47,51] and transport channels on plants [52,53], thus releasing exchangeable Cd into the soil solution, increasing soil Cd availability [54,55], and reducing crop Cd accumulation, simultaneously. This explains the result shown in Figure 3, where the Cd immobilization rate is lower than 0, even though the available Cd concentration in soil decreased with the addition of an increased dose of WS. The significant positive correlation between available Cd and available Ca (Table 4A and Figure 5), as well as the distinct negative correlation between Cd concentration in amaranth shoots and soil available Ca (Table 4B), supported the conclusions regarding the competition between Ca and Cd, both in soils and in plants. Similar results were documented by Li et al. [56], who observed little change in soil Cd mobility and a decrease in Cd concentration in rice treated with quicklime. In recent years, WS (i.e., Ca silicate) has been recognized for its strong Cd absorption capacity and has been increasingly utilized as a soil amendment for in situ remediation of contaminated soils due to its high efficiency and low cost [6,22,49]. As such, WS also holds great potential for large-scale Cd immobilization in soil.

In this study, MS was used for the first time in the remediation of Cd in contaminated soil. As an adsorption material, MS possesses an extremely strong cation exchange capacity. The NH^4+^ exchange rate of the two types of molecular sieves (4A and 13X) is much higher than that of many other adsorption materials, as well as 34 times that of the soils [57]; thereby, exchanging Cd^2+^ in soil was exchanged into MS through the ion-exchange process. It has been reported that chitosan-modified 13X MS or synthetic 13X zeolite MS can absorb heavy metals in wastewater [58]. In addition, Na^+^ in synthetic Na-A-MS may play a similar role as Ca^2+^ in the reaction with Cd [47]. This could explain why amaranth shoot Cd uptake decreased following MS application, although there was little to no change in the available Cd concentration in soil (Figure 2 and Figure 3). As shown in Appendix A, the application of MS can substantially increase the absorption of Na in amaranth shoots. This might involve either the movement of Na^+^ from MS into the soil solution through ion exchange, followed by exchange with Cd^2+^ in the soil, or increased competition between Na^+^ and Cd^2+^ on the plant’s Ca channel, resulting in an increasing Na^+^ concentration in the soil solution and a reduced concentration of available Cd upon the application of a higher amount of MS. Because of the similarity in the chemical properties of some elements, as well as their common absorption and transport mechanisms, the absorption of different elements by plants may be competitive, cooperative, or antagonistic [52]. Moreover, due to the special structure of the molecular sieve, Cd exchanged to MS by ion exchange will be more difficult to re-dissolve than Cd passivated by adsorption or precipitation, under natural conditions. MS is a potential passivation material due to its high efficiency of ion exchange and uniform microporous structure. Currently, it is rarely used in the remediation of contaminated soil but holds considerable potential for future use.

## 5. Conclusions

Suppression of Cd accumulation in amaranth was dependent on different soil properties and on the performance of immobilization agents. Other than pH, the varying immobilization performance in different soils from high (soils 2, 3, and 6) to low (soils 1, 4, and 5) suggested that OM also played an important role in affecting Cd mobility in the contaminated soils. Alkaline soil, such as soil 6, with high available Ca and OM content, as well as severe Cd contamination, was not suitable for immobilization. The application of WS and MS at a higher Si dose of 660 mg·kg^−1^ produced the optimum effect on inhibiting Cd accumulation in amaranth shoots (36% and 34%, respectively) and did not affect crop yield. Excess Ca or Na ions from WS or MS, respectively, were released into the soil solution through ion exchange, where they were either exchanged with Cd^2+^ in the soil or enhanced the competition of the plant’s Ca channel, leading to reduced Cd accumulation in plants. Thus, WS and MS can significantly inhibit the accumulation of Cd from the soil by crops and can be used to remedy farmland polluted by acidic substances and silicon-deficient heavy metals. The risk of Cd remobilization from the application of WS should be taken into consideration when establishing remediation strategies.

## Figures and Tables

**Figure 1 ijerph-18-05128-f001:**
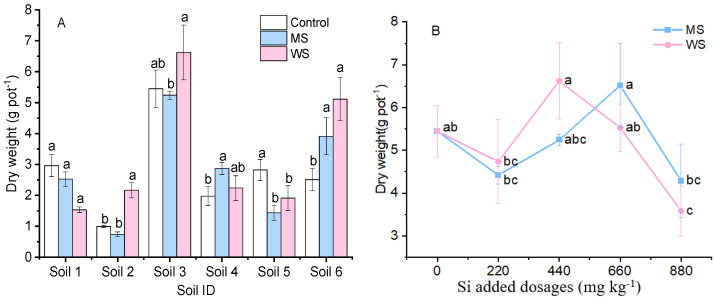
Dry weight of amaranth shoot in six Cd-contaminated soils under 440 mg·kg^−1^ Si treatments (**A**) and in soil 3 under 0, 220, 440, 660, and 880 mg·kg^−1^ Si added dosages (**B**). Control, non-amendment treatment; MS, 4A molecular sieve; WS, wollastonite. Data are means ± SE (*n* = 4). Different letters above the adjacent bars or lines denote a significant (*p* < 0.05) difference among the treatments in the same soil.

**Figure 2 ijerph-18-05128-f002:**
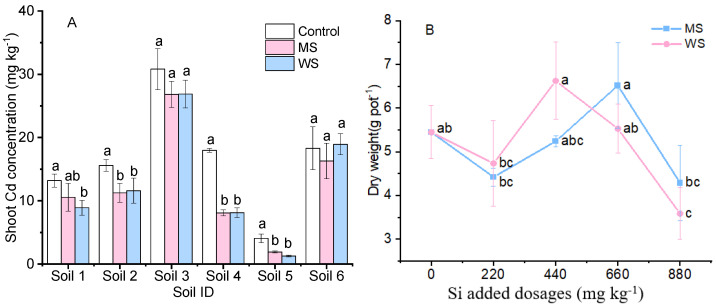
Cd concentration of amaranth shoot in six Cd-contaminated soils under 440 mg·kg^−1^ Si treatments (**A**) and in soil 3 under 0, 220, 440, 660, and 880 mg·kg^−1^ added Si dosages (**B**). Control, non-amendment treatment; MS, 4A molecular sieve; WS, wollastonite. Data are means ± SE (*n* = 4). Different letters above the adjacent bars or lines denote a significant (*p* < 0.05) difference among the treatments in the same soil.

**Figure 3 ijerph-18-05128-f003:**
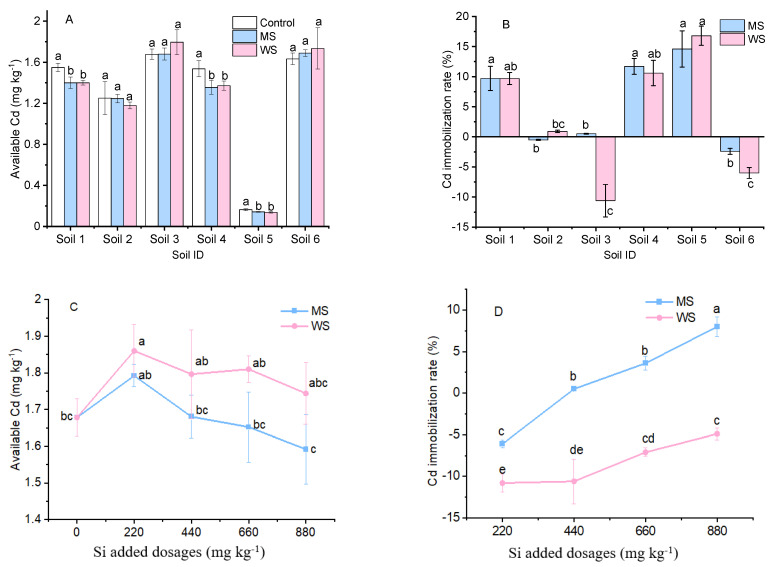
Effects of amendments on the Cd availability and Cd immobilization rate (%) in 440 mg·kg^−1^ Si amended six soils (**A**,**B**) and in soil 3 under 0, 220, 440, 660, and 880 mg·kg^−1^ Si added dosages (**C**,**D**). Control, non-amendment treatment; MS, 4A molecular sieve; WS, wollastonite. Data are means ± SE (*n* = 4). Different letters above the adjacent bars or lines denote a significant (*p* < 0.05) difference among the treatments in the same soil.

**Figure 4 ijerph-18-05128-f004:**
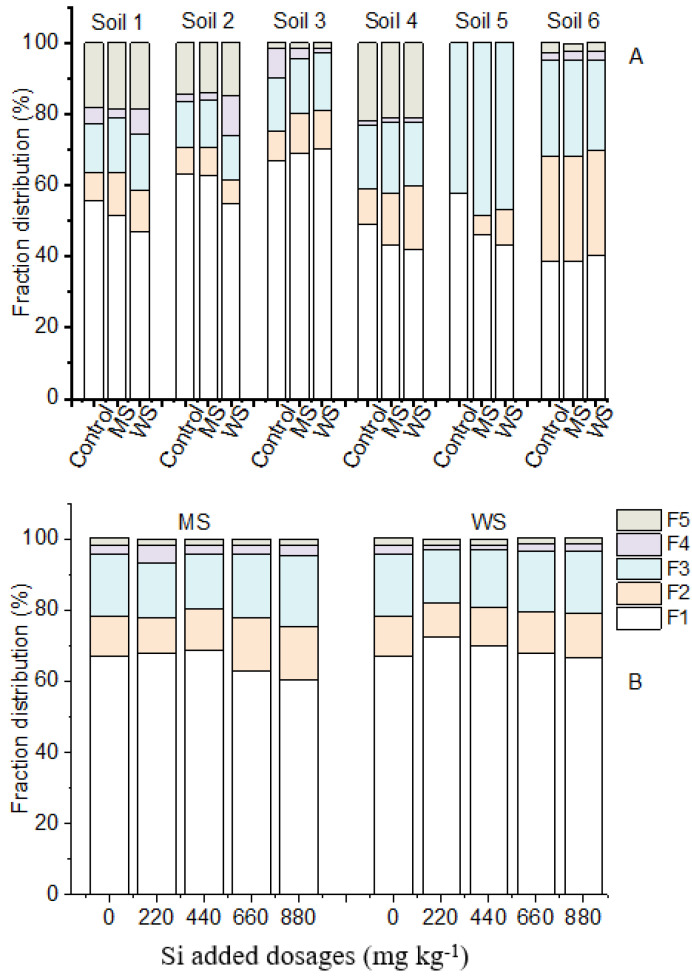
Effects of amendments on the fraction changes of Cd in 440 mg·kg^−1^ Si amended six soils (**A**) and in soil 3 under 0, 220, 440, 660, and 880 mg·kg^−1^ Si added dosages (**B**). Control, non-amendment treatment; MS, 4A molecular sieve; WS, wollastonite. F1—exchangeable fraction, F2—bound to carbonates/acid soluble fraction, F3—bound to iron and manganese oxides/reducible fraction, F4—bound to organic matter/oxidizable fraction, and F5—residual fraction.

**Figure 5 ijerph-18-05128-f005:**
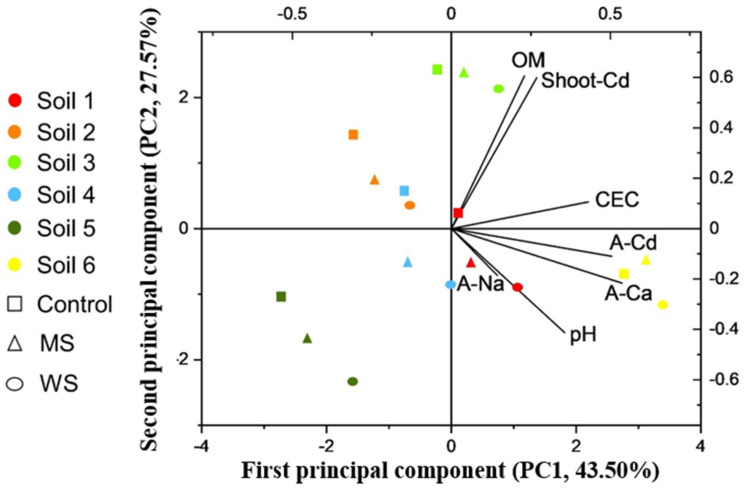
Principal component analysis (PCA) score plot of the selected soil properties and Cd accumulation dataset. Labeling corresponds to the treatments. OM: organic matter; Shoot-Cd: Cd concentration of amaranth shoot. CEC: cation exchange capacity; A-Cd/Ca/Na: soil available Cd/Ca/Na. MS, 4A molecular sieve; WS, wollastonite.

**Figure 6 ijerph-18-05128-f006:**
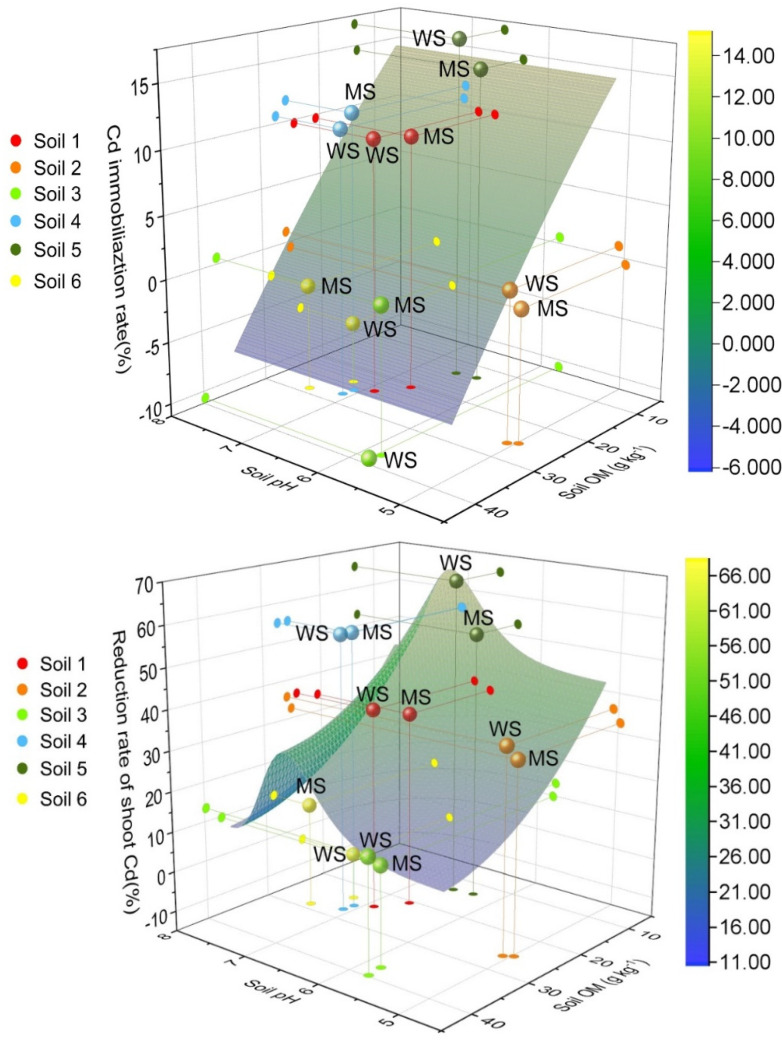
Response surfaces showing the interactive effects of soil pH and OM on Cd immobilization rate in six soils and reduction rate of shoot Cd by 4A molecular sieve and wollastonite.

**Table 1 ijerph-18-05128-t001:** Characteristics of six experimental soils.

Soils	Land Status	pH	TN	A-P	OM	CEC	A-Ca	A-K	A-Na	TCd	A-Cd
			g·kg^−1^	mg·kg^−1^	g·kg^−1^	cmol(+) kg^−1^	mg·kg^−1^	mg·kg^−1^	mg·kg^−1^	mg·kg^−1^	mg·kg^−1^
Soil 1	Farmyard close to a smelting plant	6.57	3.28	232	23.4	7.5	619	152	41.1	2.55	1.60
Soil 2	Cropland close to a smelting plant	4.90	1.80	570	31.8	2.7	136	77.8	28.9	1.69	1.21
Soil 3	Farmland around a smelting plant	5.83	1.03	502	36.2	3.9	377	126	62.7	2.97	1.87
Soil 4	Paddy around a coal-fired power plant	6.49	2.10	632	17.3	2.3	382	52.3	22.7	2.65	1.50
Soil 5	Farmland irrigated with polluted water from mine	5.92	1.45	341	9.6	0.4	219	46.4	11.0	0.38	0.15
Soil 6	Nutrition soil covering on amended mineral slag	7.77	2.50	298	39.7	10.3	1231	75.0	15.7	20.8	2.02

Note: OM: organic matter; CEC: cation exchange capacity; Total N/Cd: soil total concentration of N/Cd; A-P/Ca/K/Na/Cd: soil available P/Ca/K/Na/Cd.

**Table 2 ijerph-18-05128-t002:** Effects of adding 440 mg·kg^−1^ Si on the pH values, CEC, and concentrations of available Ca and available Na of six soils.

Soils	Treatments	pH	CEC	Ca	Na
			Cmol(+) kg^−1^	mg kg^−1^	mg kg^−1^
Soil 1	Control	5.88 ± 0.16b	16.517 ± 0.34b	564 ± 12b	34.5 ± 6.1b
Soil 2	MS	6.68 ± 0.14a	18.441 ± 0.78a	512 ± 21c	228 ± 36a
	WS	6.90 ± 0.15a	18.502 ± 0.13a	839 ± 20a	31.5 ± 5.7b
	Control	4.06 ± 0.02b	11.222 ± 0.08b	20.4 ± 0.8b	32.1 ± 1.6b
Soil 3	MS	4.98 ± 0.17a	12.729 ± 0.46a	26.2 ± 4.5b	239 ± 43a
	WS	5.07 ± 0.08a	13.108 ± 0.21a	221 ± 15a	39.5 ± 1.6b
	Control	5.40 ± 0.12b	9.9 ± 0.28c	207 ± 19b	42.4 ± 5.2b
Soil 4	MS	5.87 ± 0.13a	12.821 ± 0.38a	149 ± 12c	251 ± 25.7a
	WS	5.77 ± 0.17a	12.180 ± 0.26b	434 ± 35a	34.8 ± 3.3b
	Control	6.21 ± 0.07b	7.6 ± 0.3c	215 ± 20b	20.2 ± 3.5b
Soil 5	MS	7.08 ± 0.12a	9.0 ± 0.42b	205 ± 23b	248 ± 40a
	WS	7.10 ± 0.14a	9.8 ± 0.29a	442 ± 44a	22.9 ± 5b
	Control	5.28 ± 0.19b	2.8 ± 0.4b	23.5 ± 2.6b	23.4 ± 4.6b
Soil 6	MS	6.35 ± 0.11a	4.9 ± 0.44ab	22.2 ± 1.6b	288 ± 20.7a
	WS	6.57 ± 0.08a	5.7 ± 0.17a	274 ± 39a	26.3 ± 1.6b
	Control	7.24 ± 0.19a	15.354 ± 1.79a	1433 ± 47ab	34.66 ± 4.9b
	MS	7.47 ± 0.15a	16.900 ± 0.24a	1395 ± 32b	307 ± 30a
	WS	7.24 ± 0.16a	15.568 ± 0.89a	1537 ± 97a	39 ± 7b

Control, non-amendment control; MS, 4A molecular sieve; and WS, wollastonite. Data are means ± SE (*n* = 4). Same letters beside the means indicate no significant difference according to Tukey honestly significant difference test (*p* < 0.05).

**Table 3 ijerph-18-05128-t003:** Effects of different added dosages of Si on the pH values, CEC, and concentrations of available Ca and available Na of soil 3.

Treatments	mg kg^−1^	pH	CEC	Ca	Na
			Cmol(+) kg^−1^	mg kg^−1^	mg kg^−1^
Control	0	5.40 ± 0.12d	9.9 ± 0.3c	207 ± 19de	42.4 ± 5e
MS	220	5.65 ± 0.14cd	12.5 ± 0.3a	169 ± 11de	177 ± 17.7d
	440	5.87 ± 0.13bc	12.8 ± 0.4a	149 ± 12e	251 ± 25.7c
	660	6.83 ± 0.19a	12.8 ± 0.2a	212 ± 19de	349 ± 18b
	880	6.66 ± 0.15a	12.7 ± 0.4a	150 ± 26e	452 ± 19.5a
WS	220	5.43 ± 0.18d	11.577 ± 0.5b	225 ± 50d	36 ± 8e
	440	5.77 ± 0.17bc	12.180 ± 0.3ab	434 ± 35c	40.4 ± 3.3e
	660	6.06 ± 0.11b	12.468 ± 0.1ab	595 ± 37b	34.8 ± 3.2e
	880	6.55 ± 0.04a	13.341 ± 0.1a	706 ± 40a	32.7 ± 4.6e

Control, non-amendment control; MS, 4A molecular sieve; and WS, wollastonite. Data are means ± SE (*n* = 4). Same letters beside the means indicate no significant difference according to Tukey honestly significant difference test (*p* < 0.05).

**Table 4 ijerph-18-05128-t004:** Pearson’s correlation coefficients among soil chemistries, soil Cd availability, and Cd concentrations in plants.

	pH	OM	CEC	A-P	A-Cd	A-Ca	A-Na
A: correlation matrix of six soils under 440 mg·kg^−1^ Si added treatments							
shoot-Cd	−0.175	0.825 **	0.314 **	0.186	0.768 **	0.182	−0.305 *
A-Cd	0.170	0.775 **	0.708 **	0.299 **	1	0.473 **	−0.132
B: correlation matrix of soil 3 under 0, 220, 440, 660 and 880 mg·kg^−1^ Si added dosages							
shoot-Cd	−0.853 **	−0.120	−0.268	−0.035	0.311	−0.462 **	−0.327
A-Cd	−0.423 **	0.402 **	0.061	0.080	1	0.279	−0.547 **

Note: shoot-Cd: Cd concentration of amaranth shoot; OM: organic matter; CEC: cation exchange capacity; A-P/Cd/Ca/Na: soil available P/Cd/Ca/Na. * *p* < 0.05; ** *p* < 0.01.

## Data Availability

Data supporting reported results can be found in the Appendix A.

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
