# Peer review of "Immobilization of Cadmium by Molecular Sieve and Wollastonite Is Soil pH and Organic Matter Dependent"

_ijerph, 2021, doi:10.3390/ijerph18105128_

Round 1
Reviewer 1 Report
Introduction:
- Line#27: “Rapid economic..” : The sentence is vague. Needs clarification.
- The novelty of the work needs to be mentioned clearly.
- Line #68, Paragraph “In our previous studies..”: More description of the material MS and WS is needed.
- Why MS and WS is chosen for immobilization of Cd: needs further clarification.
- Why Amaranth has been chosen as the vegetation under study?
Materials and Methods:
- Line #109: “Twenty-four treatments..” : 1.9 g/Kg and 2.08 g/Kg are dosages of WS and MS, not their rates. Please correct
- Line #141: “The CEC of soils..” Further description of the method is needed.
Result:
- Line #191: Is it 118%? Please confirm.
- Section “Amaranth biomass and Cd accumulation”: The result from the graph is described nicely. However, it is necessary to illustrate the significance of those numbers and percentages here. Also, it is necessary to mention few previous references to support your result.
- Line #229-234: You need to provide few explanations here for these observations with some background references.
- Line #238-243: You need to provide few explanations here for these observations with some background references.
- Section 3.3: Please provide explanation for the low immobilization rate? Are there any previous reference for this?
- Line #278-281: What is the reason for the increased amount of exchangeable Cd? In this case, what is the significance of this treatment? Please explain.
- Line #286: Same comment as #6.
Discussion:
- The discussion could have been given with the consecutive result. It would make the discussion more relatable and easier to understand.
Reviewer 2 Report
The work presents a beautiful study on the adsorption of Cd in contaminated soils through the use of molecular sieves and wollastonite. The work is very well presented, outlined and discussed, presenting all the necessary controls and statistical precision. The quality of the figures' legends is impressive. For these reasons, I suggest your approval for publication. Some considerations:
- check the form of presentation of equations, according to the guide for authors;
- In the abstract, the sentence "No significant difference was observed between the effects of both MS and WS on Cd accumulation" suggests that the application of MS w WS was not significant. Please review;
- authors can provide more details of the adsorbent materials used, such as trademark, surface area, pore distribution and chemical composition. This information can complement the supplementary material already presented;
- It is not recommended to vary the units and measures, which, when composed, must be presented by juxtaposition or separated by point or space. I suggest to the authors to change the presentations of the units, such as: "A low amendment dose (220 mg Si kg-1)" to "A low Si dose (220 mg kg-1)";
- Table 1 can be seen in the material section. Please check your note;
